# Importance of Knife Sharpness during Slaughter: Shariah and Kosher Perspective and Scientific Validation

**DOI:** 10.3390/ani13111751

**Published:** 2023-05-25

**Authors:** Pavan Kumar, Ahmed Abubakar Abubakar, Jurhamid Columbres Imlan, Muideen Adewale Ahmed, Yong-Meng Goh, Ubedullah Kaka, Zulkifli Idrus, Awis Qurni Sazili

**Affiliations:** 1Institute of Tropical Agriculture and Food Security, Universiti Putra Malaysia, UPM, Serdang 43400, Selangor, Malaysia; pavankumar@gadvasu.in (P.K.); ahmadsadeeq7@gmail.com (A.A.A.); muideenahmed13@gmail.com (M.A.A.); zulidrus@upm.edu.my (Z.I.); 2Department of Livestock Products Technology, College of Veterinary Science, Guru Angad Dev Veterinary and Animal Sciences University, Ludhiana 141004, Punjab, India; 3Department of Animal Science, College of Agriculture, University of Southern Mindanao, Cotabato 9407, Philippines; jurhamidimlan@yahoo.com.ph; 4Department of Preclinical Sciences, Faculty of Veterinary Medicine, Universiti Putra Malaysia, UPM, Serdang 43400, Selangor, Malaysia; ymgoh@upm.edu.my; 5Department of Companion Animal Medicine and Surgery, Faculty of Veterinary Medicine, Universiti Putra Malaysia, UPM, Serdang 43400, Selangor, Malaysia; dr_ubedkaka@upm.edu.my; 6Halal Products Research Institute, Universiti Putra Malaysia, Putra Infoport, UPM, Serdang 43400, Selangor, Malaysia; 7Department of Animal Science, Faculty of Agriculture, Universiti Putra Malaysia, UPM, Serdang 43400, Selangor, Malaysia

**Keywords:** religious slaughter, knife sharpness, neck cut position, slaughter skill, restraint, animal welfare

## Abstract

**Simple Summary:**

A sharp knife of appropriate dimension (blade length) is very important during halal and kosher slaughtering of animals without stunning for a rapid and clean neck severance. It improves bleeding and alleviates pain and stress in animals by early onset of unconsciousness. An efficient bleed-out improves meat quality and food safety. With the ever-increasing demand for halal and kosher meat due to its awareness, authenticity, nutritive value, and animal welfare compliance, there is an urgent need to emphasize the role of knife sharpness during slaughter as per the prescribed religious practices. Other issues such as neck cut positions, blade length of the knife, proper training of slaughterhouse workers, infrastructure, and constant monitoring of the slaughtering process also need to be addressed to improve animal welfare and meat quality.

**Abstract:**

Halal and kosher slaughter have given the utmost importance to the sharpness of knives during the slaughter of animals. A sharp knife of appropriate dimension (blade length) makes slaughter less painful during neck severance and facilitates desirable bleeding. The role of knife sharpness has not been given due credit from an animal welfare perspective and is likely ignored by the people involved in slaughterhouses. A neat, clean, and efficient neck cut by an extremely sharp knife reduces the pain. It improves the bleeding out, thus making animals unconscious early without undergoing unnecessary pain and stress. It also helps in improving meat quality and food safety. A slight incremental improvement in knife sharpness could significantly improve the animal welfare, productivity, efficiency, and safety of meat plant workers. The present review critically analyzed the significance of knife sharpness in religious slaughter by reducing stress and pain and improving meat quality and food safety. The objective quantification of knife sharpness, proper regular training of slaughterers, and slow slaughter rate are the challenges faced by the meat industry.

## 1. Introduction

The World Organization for Animal Health (WOAH) defines animal welfare as *“the physical and mental state of an animal concerning the condition in which it lives and dies”* [1]. It consists of the responses of an animal to socio-physiological factors, escapes or avoidance behavior, animal physiology, and biochemistry [2]. Each society has its interpretation of animal welfare based on its moral and ethical values and welfare standards [3]. With increasing awareness, and education, consumers prefer food not only for good nutritive quality, sustainability, and processing but also for ethical and spiritual quality. Consumers are more concerned about handling animals and management practices involved during meat production, such as stunning prior to slaughter, and free-range chicken production. The animal welfare issue has taken center stage in global meat production and marketing [4,5,6,7,8,9]. Further, a well-established link between pain or stress during slaughter and its negative impact on meat quality warrants the immediate attention of researchers and policy-makers for proper animal welfare compliance during animal slaughtering [3]. 

The pre-slaughter handling and slaughter process is crucial from an animal welfare perspective as it comprises converting the live animal into edible pieces of meat for human consumption. Several pre-slaughter factors cause an animal stress, fear, pain, and distress, thus compromising their welfare [10,11]. Pain is “*an unpleasant sensory and emotional experience associated with or resembling real or potential tissue injury*” [12]. Stress is a complex physiological state that comprises a range of integrative and behavioral changes in response to a real or perceived threat to homeostasis. In contrast, fear denotes a condition of danger perception or potential harm that could compromise an animal’s safety [13]. Distress denotes a negative and aversive state under which the ability of an animal to cope and adapt is impaired [14]. Pre-slaughter handling practices have a significant effect on animal stress, pain, distress, and fear, such as transportation conditions, loading and unloading, stocking density, water and feed availability, attitude and training of animal handlers, and slaughtering practices (such as stunning, restraints, knife sharpness, and training of butchers). Knife characteristics, especially knife sharpness, are very crucial among these factors. 

During kosher and halal slaughter, animals should be restrained by using minimum stressful methods to hold the animals for neck cuts. Gentle handling and less stressful restraints lower the issue of delayed periods of consciousness after a neck cut and improve animal welfare [15]. These considerations also affect the reaction of the animal to a throat cut [16,17,18], the issue of prolonged consciousness after the neck cut [19], and the perfusion of blood in the respiratory tract [20]. Tight/robust restraints cause stress—especially in cases where animals are turned on their side or back in a rotatory casting pen—substantial tissue damage, and prolonged time to reach the stage of unconsciousness, thereby feeling pain, distress, anxiety, and suffering [21]. The issue of gentle handling and less stressful restraining methods are more critical in the slaughter of cattle without stunning as compared to sheep. This could be due to the larger body size of cattle [15] and anatomical differences in the blood supply to the brain; thereby taking longer time to reach the stage of unconsciousness in cattle as compared to sheep [22,23]. The World Organization for Animal Health (WOAH, previously known as OIE) recognized fully inverted, upright, and lateral/sideway restraints for animals, with suspending only allowed in poultry [24]. 

Even though religious authorities (both Islam and Judaism) and sacred texts emphasize the importance of using a sharp knife and maintaining its sharpness during slaughter, there is very little scientific evidence available to corroborate this element, especially in the context of ritual slaughter (halal and kosher slaughter). Religious slaughter implies the slaughter of permissible animals by severing the trachea, esophagus, and blood vessels using a sharp blade and following laws as prescribed by rituals (blessings/invocations) that characterize its purity/ethical value [25]. Islam and Judaism have given utmost importance and zero tolerance towards animal welfare compliance during slaughter and strongly advocate benevolent and compassionate treatment of animals. 

A clean neck cut at the proper position using a very sharp knife facilitates rapid blood loss and fewer incidences of false aneurysm formation, thereby leading to the early onset of unconsciousness. Specific details about sharpness and its dimensions for the *Chalaf* are mentioned for kosher slaughter. However, this very critical factor is largely overlooked in scientific literature. In this context, this paper critically reviews the significance of knife sharpness in religious animal slaughter without stunning under the broad ambit of animal welfare principles by alleviating stress and pain, this ultimately improving meat quality. The manuscript also reviewed various factors that could render knife sharpness as an important determinant of animal welfare during slaughter such as the position of the neck cut, knife dimension, training/expertise of the slaughterman, restraints, and presentation of the animal during slaughter. 

## 2. Sharia’s Perspective of Knife Sharpness

A sharp knife is recommended for halal and kosher slaughter for efficient bleeding, alleviating pain, and producing quality meat [26,27]. Islam prescribes proper guidelines for ensuring the sharpness of knives and the slaughter of an animal by complying with animal welfare principles mentioned in the Holy Quran and Hadiths. 

“Certainly Allah has decreed proficiency in all things. Thus … if you perform slaughter (zabh), perform it well (painlessly). Let each of you sharpen his knife/blade and let him minimize suffering to the animal he slaughters (zabiha die painlessly/peacefully).”

“Allah calls for mercy in everything, so be merciful when you kill and when you slaughter; sharpen your blade to relieve its pain.” (Al-Qaradawi, 1994).

“Allah has commanded you to treat all creatures with kindness. When you slaughter an animal, do so kindly. Sharpen the knife well and give comfort to the animal being slaughtered.” 

The Islamic tradition strongly advocates the humane slaughter of animals to alleviate pain and suffering. For example, *Sahih Muslim (Book 21, Chapter 11, Number 4810)* records Prophet Mohammad (PBUH) saying:

“Verily Allah has enjoined goodness to everything; so when you kill, kill in a good way, and when you slaughter, slaughter in a good way. So every one of you should sharpen his knife and let the slaughtered animal die comfortably.”

“When one of you slaughters, let him complete it.”

حَدَّثَنَا أَبُو بَكْرِ بْنُ أَبِي شَيْبَةَ، حَدَّثَنَا إِسْمَاعِيلُ ابْنُ عُلَيَّةَ، عَنْ خَالِدٍ الْحَذَّاءِ، عَنْ أَبِي قِلاَبَةَ، عَنْ أَبِي الأَشْعَثِ، عَنْ شَدَّادِ بْنِ أَوْسٍ، قَالَ ثِنْتَانِ حَفِظْتُهُمَا عَنْ رَسُولِ اللَّهِ صلى الله عليه وسلم قَالَ “ إِنَّ اللَّهَ كَتَبَ الإِحْسَانَ عَلَى كُلِّ شَىْءٍ فَإِذَا قَتَلْتُمْ فَأَحْسِنُوا الْقِتْلَةَ وَإِذَا ذَبَحْتُمْ فَأَحْسِنُوا الذَّبْحَ وَلْيُحِدَّ أَحَدُكُمْ شَفْرَتَهُ فَلْيُرِحْ ذَبِيحَتَهُ ”

In accordance with the Prophet’s (PBUH) commandment that at the point of slaughter, the knife’s cutting edge must be well sharpened (Jama’ulFawa’id). In Islam, using bones, claws, teeth, nails, and alike is strictly forbidden. This implies less pain to the animals during slaughter. 

“Use everything to slaughter which allows blood to flow, except for teeth and nails, and all else is permissible” (Sahih Bukhari, p 827; Sunan Abu Dawood). 

حَدَّثَنَا عَمْرُو بْنُ عَلِيٍّ، حَدَّثَنَا يَحْيَى، حَدَّثَنَا سُفْيَانُ، حَدَّثَنَا أَبِي، عَنْ عَبَايَةَ بْنِ رِفَاعَةَ بْنِ رَافِعِ بْنِ خَدِيجٍ، عَنْ رَافِعِ بْنِ خَدِيجٍ، قَالَ قُلْتُ يَا رَسُولَ اللَّهِ إِنَّا لاَقُو الْعَدُوِّ غَدًا، وَلَيْسَتْ مَعَنَا مُدًى فَقَالَ ” اعْجَلْ أَوْ أَرِنْ مَا أَنْهَرَ الدَّمَ وَذُكِرَ اسْمُ اللَّهِ فَكُلْ، لَيْسَ السِّنَّ وَالظُّفُرَ، وَسَأُحَدِّثُكَ، أَمَّا السِّنُّ فَعَظْمٌ، وَأَمَّا الظُّفُرُ فَمُدَى الْحَبَشَةِ ” وَأَصَبْنَا نَهْبَ إِبِلٍ وَغَنَمٍ فَنَدَّ مِنْهَا بَعِيرٌ، فَرَمَاهُ رَجُلٌ بِسَهْمٍ فَحَبَسَهُ فَقَالَ رَسُولُ اللَّهِ صلى الله عليه وسلم ” إِنَّ لِهَذِهِ الإِبِلِ أَوَابِدَ كَأَوَابِدِ الْوَحْشِ، فَإِذَا غَلَبَكُمْ مِنْهَا شَىْءٌ، فَافْعَلُوا بِهِ هَكَذَا, 

The incision should be instantaneous, with one uniform directional movement, and achieved without interruptions, uncertainty, or force [28]. 

Furthermore, the cut must be made from the ventral position of the neck near the lower jaw and just before the spine [28]. This is following hadith that “*the jugular veins and the carotid arteries (wadaja’an), in addition, to the throat (hulqum) and the trachea (mari’), but the vertebral or spinal cord must not be cut.”* The head must not be wholly separated from the remaining body during slaughter. 

“The knife must be razor sharp and without blemishes and damage. For animals with normal necks, the slaughter must begin with an incision on the neck just before the glottis, and for animals with long necks, such as chickens, turkeys, ostriches, camels, etc., the incision must be before the glottis”.

“…must be done once only. The slaughtering implements must not be lifted off the animal during slaughtering. Any lifting is construed as one act of slaughter. Multiple acts of slaughter on one animal are prohibited”.

Under Halal, acceptable animals and birds are slaughtered with a razor-sharp knife to have a swift, deep incision cutting the front of the esophagus, trachea, jugular veins, and carotid arteries [29]. In Halal slaughter, proper emphasis is given to the knife’s sharpness to facilitate rapid and efficient blood drain and the onset of unconsciousness. 

## 3. The Kosher Perspective of Knife Sharpness

Kosher slaughter/*Shechitah*/*Shechita* is the animal slaughter method followed by the Jewish community derived from a *mitzvà* (commandment) mentioned in the book of Deuteronomy 12:21

“And ye shall be men of holy calling unto Me, and ye shall not eat any meat that is torn in the field” (Exodus XXII:30)

“...thou shall kill of thy herd and of thy flocks, which the Lord hath given thee, as I have commanded thee...” (Deuteronomy XII:21). “[...] you may slaughter animals from the herds and flocks the Lord has given you, as I have commanded you, and in your towns, you may eat as much of them as you want”.

Rapid blood loss and maximum bleed-out is recommended in Kosher slaughter as 

“Only be sure that thou eat not the blood: for the blood is life” (Deuteronomy 12:23)

In Kosher meat production, three factors, viz., permitted animals, strict prohibition of blood, and mixing meat with milk, are vital considerations [30]. Wild birds and pigs are considered impure in kosher diets [31]. In Jewish law (*Halacha*), importance is given to compliance with animal welfare and avoiding pain and stress during slaughtering (Ha Levi A. 13th cent, Karo 1563d) [30]. During Shechitah, Jewish law (*Halacha*) emphasizes the suitability of a knife (sharpness and size), immobilization of animals, and neck cut (correct knowledge of anatomy and physiology, and skill [32]. 

In kosher slaughter, the knife’s sharpness, its absence of nicks, a pre-slaughter examination of the knife’s sharpness, and its size are all given careful consideration (double the width of the neck of the animal to be slaughtered). It is essential for decreasing stress and suffering during animal slaughter [33]. The sharpness and size of the knife (*Chalaf,* also known as *Chalef/Chalof/Chalif*), along with the rigorous training and inspection by Jewish authorities, ensure a proper supply of kosher meat and its production [34].

Kosher slaughter is completed in five phases, viz., selection of animal (cloven hooves and ability to ruminate as per The New International Version of the Bible 2011, Leviticus XI), health inspection of the animal, slaughter, inspection (*bedika*), and cleaning (*nikkur*) of meat, and koshering (washing and salting meat for removing blood) [27]. *Shechitah* is performed by making a clean incision at the front of the neck and cutting the trachea, esophagus, carotid arteries, and jugular veins by using a *Shechitah* knife (*Chalaf*, derived from the Hebrew word meaning ‘to change’) with a *Shochet* (authorized slaughter man) as per fundamental commandment, conveyed via the Oral Law and dating back to the time of Moses [18]. 

The *Chalaf* is designed to have exquisite sharpness and is repeatedly inspected between animals to avoid imperfections [18]. There are five principles of halachic (traditional body of Jewish law) during the *Shechitah* viz. *Shehiyah*/pause (uninterrupted incision), *Derasah*/pressure (no pressing of the blade against the neck), *Halad*/stabbing (adequate length of the blade so it does not get covered with wool, feathers, or hide), *Hagramah*/slanting (severing neck at the appropriate point for neat, clean, and efficient incision), and *Ikkur*/tearing (no tearing of tissues) [18,35]. An animal is deemed unfit if any problem is found with the knife or the neck cut [36]. 

*Chalaf* should be perfectly smooth and razor-sharp without any nicks or serrations to facilitate the slaughter process as painlessly as possible. It is twice the length of the neck of the animals going to be slaughtered (poultry: 14–16 cm, sheep and goats: 25 cm, cattle: 40–45 cm) [37]. Before and after *Shechitah*, *chalaf* must be inspected along with fingernails to ensure compliance with Jewish slaughter regulations during the cut, and any presence of a nick makes the meat unfit for consumption (*terefa/terefah*) or rejection [34]. In chicken slaughter, the *Chalaf* may be inspected following the killing of all animals in a group and checked for nicks, with the provision that all slaughtered birds are considered *terefah*/rejected if nicks are discovered [38]. 

Further, detecting a nick is tricky as a trained shochet may detect a nick that generally goes unnoticed by the trained sharpener [38]. It is mandatory to check the neck of animals to ensure the absence of any materials that may damage the knife. In case of potential knife-damaging materials such as dirt, dust, soil, etc., the animal should be washed. This process during kosher slaughter slows down the slaughtering process in addition to being a cause of stress. In religious slaughter, washing animals while entering the lairage is widely followed so that animals will have sufficient time to recover from stress [38]. However, proper care should be taken in cold climatic conditions, as it may lead to severe cold stress in animals.

## 4. Knife Sharpness in Animal Welfare during Slaughter

Sharpness significantly affected the forces generated and energy required during the cutting process, the cutting-edge durability, and the surface finish [39]. In the USA, Section 1902 of the Humane Methods of Livestock Slaughter Act of 1978 stipulates religious slaughter by a method in which “*the animal undergoes unconsciousness by the lack of blood to the brain due to the instantaneous severing of carotid arteries with a sharp instrument and proper handling in accordance with such slaughter*” [40]. 

The knife design and neck cut process were crucial in preventing the animal from reacting to the cut, ensuring rapid blood flow [28,41]. The knife’s sharpness and a clean and uninterrupted cut also influence vasoconstriction, clotting, and ballooning resulting in carotid occlusion/false aneurysm [42] due to constriction of the caudal end of the severed carotid arteries. While the sharpness of the knife influences neck cutting, the method of neck cutting and the number of cuts influence pain perception [19]. A rapid and efficient cut by a razor-sharp knife is crucial for maximum blood loss and early onset of a state of unconsciousness, thus reducing the pain during the whole process. In addition to this, it also improves meat quality by reducing petechial hemorrhages. Stressed and excited animals usually take more time to become unconscious than calm animals, thus not meeting the higher animal welfare standards [9,32]. 

To preserve animal welfare, religious slaughter without stunning requires improved administration. The animal remains conscious during the neck cut, and it will take some time (varies with species or neck cut) to undergo a state of unconsciousness. Thus, the animal remains sensitive to pain and stress between this neck cut and unconsciousness. It was observed that if religious slaughter without stunning was performed with a proper razor-sharp knife on a calm and rested animal, restrained properly/comfortably, then animals showed very little (flinch) or no reaction to the neck cut [43]. This reaction/flinch was noticed even less than an ear tag punch, a metal clanging noise, air hissing, followed by no further reaction afterwards [43]. Similarly, Barnett et al. [44] also observed mild physical response to neck cutting in some birds (100 birds) whereas no response was observed in the majority of birds (592 birds) during kosher slaughter. The authors [44] also observed the presence of physical response to touching the eye or eyelid in birds up to 5 s after a neck cut, which disappeared after 15 s of the neck cut. The loss of posture and presence of involuntary muscular contraction was noticed after 12–15 s of neck cutting and the loss of 40% of total blood within 30 s of neck severance. However, some animal welfare scientists believe that the pain during religious slaughter could be reduced if religious slaughter is performed correctly [45]. Further, low behavioral responses after a neck cut may not necessarily indicate the pain-free status of the animal [45,46].

As blood is an excellent medium for the growth of various microorganisms, maximum possible drainage is recommended for improving the quality of meat. Further, the presence of blood in carcasses could also make the appearance of the carcass dark, which consumers prefer less. Kosherization has also been reported to improve meat quality by reducing *Salmonella* and coliform counts [47]. The hemoglobin in the blood is a potent lipid prooxidant, and efficient bleeding is also recommended to improve the meat’s oxidative stability. Further slaughter and bleeding methods have been reported to affect meat color and sensory properties [48,49]. Koshering (slating and washing) also has an effect on meat quality by significantly reducing shear force and drip loss [50]. 

### 4.1. Knife Sharpness on Pain Sensation

Very few studies have evaluated the effect on animal welfare and meat quality associated with the sharpness of slaughter knives [51]. If restrained properly without pressure and gently done, cattle were observed to have little or no reaction to the throat cut in three kosher slaughterhouses [43]. When the blade touched the skin, cattle showed a slight flinch which was less vigorous than the animals’ reactions to an ear tag, and cattle were observed to remain calm as the cut proceeded [43]. Further, the wound should be open during the incision to prevent pain. Further, the knife should be of sufficient length, so that its tip remains outside the neck during the cut [52]. 

The presence of nicks on blades is considered to cause irritation while cutting blood vessels and associated tissues during halal and kosher slaughter. Gibson et al. [16,53,54,55,56,57] noted that the electroencephalogram spectrum varies with neck cutting and could be correlated with pain sensation. However, the size of the knife (10 inches) and its machine sharpening used in the study make it difficult to conclude pain and stress in animals during religious slaughter, particularly kosher slaughter [34,38]. Further, the short knife even sharpened on a grindstone used in these studies could cause the tip to gouge the throat [52]. 

### 4.2. Sharpness and Blade Length

Sharp knives need a lower force requirement than blunt knives. This lower brute force lowers the damage to the meat. Knife sharpness is correlated with the forces produced and energy needed during the cutting process, the cutting edge, and the cutting surface [39]. Grandin [58] observed that a short knife used in the halal slaughter of cattle indicates that digging the end of the knife blade into the throat caused intense reaction and pain. Muslim scholars recommended a sufficient length of knife used in Halal slaughter with a minimum of 18 cm for cows and 24 cm for buffaloes. Likewise, Velarde et al. [59] observed that the blade length used for halal slaughter was 29.6 ± 1.79 cm, whereas size and shape were more uniform for kosher slaughter (approx. 40 cm *Chalaf* knife). The blade length of knives used for the halal slaughter of sheep (without stunning) was 22.2 ± 1.82 cm and 25 cm in case of kosher slaughter [59]. Table 1 summarizes the blade length of knives used in halal and kosher slaughtering. 

Perfect sharpness and proper height of the knife ensure the severing of all jugular and carotids with a rapid stroke resulting in a sudden drop of arterial pressure to the brain and fast and massive blood loss [18]. In a survey of slaughterhouses in Italy regarding kosher slaughter practices, Bozzo et al. [35] observed the majority of rejection (2.4% of total samples) due to failure to complywith a pause (*Shehiyah*) followed by non-compliance to the pressure (*Derasah*) and stabbing *(Halad)* rule. The authors also did not notice any rejection due to non-compliance to slanting and tearing. This further strengthens the requirement for sharp knives of the appropriate dimension for kosher slaughter. 

In addition, Abd El-Rahim [61] advocated that the minimum length of the sharp edge of a knife used to slaughter animals should be at least 12 cm. Further elaborating, Helmut [28] recommended that the knife be constantly sharpened and free from scratches and faults as it may cause pain by dragging and grasping the tissues during a neck cut. According to personal observation by Leffert [45], the blades used for religious slaughter in the USA were of 6–18 inch size with significant variations in the degree of sharpness [45]. This could have affected the variation in the onset of unconsciousness in animals slaughtered by religious methods from 10–60 s [45] in the slaughterhouses visited by the author in the USA. 

The shorter knife takes more time to perform neck cuts as its tip may gouge the wound and become stuck in the cut, with sawing motion during the cutting procedure. This would stimulate accessible nerve endings in the skin and at the cut edge of the skin leading to potential pain perception [19]. There have been reports of using blunt knives in halal slaughter in various countries [19,62], consequently requiring more pressure and more attempts during neck incisions, potentially provoking pain. In water buffalo, in lateral recumbency, neck twisting is impossible due to the horns’ breadth. In such cases, a blade is inserted into the skin, and after the animal has settled down, the blade is repositioned in the wound, and several vertical cuts (up to 7–19 cuts) are made, as observed by Gregory et al. [20]. There is a need for specific legislation, guidelines, and training programs for handlers to improve the welfare of water buffaloes during slaughter [63].

### 4.3. Sharpness on Blood Biochemical and Electroencephalogram (EEG)

Imlan et al. [51] evaluated the effect of knife sharpness on stress and pain perfection in Brahman crossbred steers in terms of blood enzymes, plasma catecholamines, and electroencephalogram (EEG) changes. The authors reported a significant difference between the concentration of pre- and post-slaughter glucose, creatine kinase (CK), and lactate dehydrogenase (LDH) concentration in steers slaughtered with a sharp commercial knife (ANAGO sharpness score of 7.8) commonly used in slaughterhouses as compared to those slaughtered with a sharp knife (sharpened with a machine with ANAGO sharpness score of 8.0) [51]. Further, the authors observed a significant increase (*p* < 0.0001) in catecholamines (adrenaline), glucose, CK, and LDH post-slaughter in comparison to pre-slaughter in the commercial sharp knife group as compared to the sharp knife group. Increased LDH and CK in the blood indicate muscle damage, fatigue, and stress in animals [4,6,64,65]. Similarly, the release of catecholamines (epinephrine/adrenaline and nor-epinephrine/nor-adrenaline) in the blood indicates an initial reaction to stress/fear [17,66]. 

An electroencephalogram presents the electrical activity of neurons rapidly, accurately, and objectively [67,68]. These variables are well interrelated with animals’ physiological and biochemical parameters [3]. On analyzing electroencephalogram (EEG) variables, a significant increase was noted in the median frequency (F_50_) (*p* < 0.0001) and total power (Ptot) (*p* < 0.0001) in the animals slaughtered with the sharp commercial knife as compared to those slaughtered with the sharp knife [51]. These parameters could be correlated with pain and stress observed during neck cuts [17,57,69,70]. The significantly increased alpha, beta, delta, and theta waves in the EEG spectrum in cattle after slaughter increased could be attributed to use of razor-sharp knives resulting in a mild behavioral response in animals being non-painful to the animal [38]. 

A sharp change in the EEG spectrum upon neck cutting is depicted in Figure 1. 

## 5. Mechanical Attributes of the Knife 

In general, knife sharpness refers to the design, fitness, and quality of cutting-edge design and is measured in terms of the forces needed for cutting [71]. Sharpness can be defined by the level of force exerted by the knife during the cutting of the material [72] or the area of the cutting edge [73]. The main parameters determining the efficiency of all cutting operations depend on the durability, thickness, consistency, and speed accomplished by blades and correlate to the suitable application. In various meat-cutting operations, professional workers have duly acknowledged that the knife blade’s sharpness affects the workers’ productivity and the product’s quality [72]. Further, the grip forces and cutting actions were greater with blunt but workable knife blades vs. recently sharpened knives [72] with a 20–30% decline in grip force, cutting actions, and time with sharper contrast with blunter blades. 

It is highly preferable to use a sharp knife with a durable edge. The sharpness of the blade is determined by several factors, such as the properties of the steel, the relative movements of the blade and target material, the curvature and edge angle of the blade, grinding, and finish [74]. A good sharpness condition needs a cutting force of 25 N and a blunt condition with a 75 N cutting force. Moreover, McGorry et al. [74] concluded that while a sharp blade finish lowers the force needed for elongated cuts through different tissues, remarkably, the finish has a comparable effect for shorter cuts through muscle only as a tendency. 

## 6. Quantifying the Sharpness

The objective quantification of the sharpness of the knife is quite challenging. During religious slaughter, it is carried out by the slaughterman prior to religious slaughter. It is also recommended to sharpen the knife and inspect its sharpness before halal or kosher slaughter by Muslims and Jewish authorities. As sharpness is directly linked with animal welfare compliance and meat quality, there is, thus, an urgent need to objectively quantify the sharpness of knives used in religious slaughter. The objective quantification of the sharpness of the knife is quite challenging. During religious slaughter, it is carried out by the slaughterman prior to the religious slaughter. Thus, it is quite challenging to standardize the sharpness measurement under various slaughtering conditions. The current measurement of knife sharpness is basically done by visual inspection/assessment, which is very subjective and varies with individual experience, training, and expertise. Commercial slaughterhouses have very high slaughter rates, so the step of knife sharpness assessment and maintaining the desired level of knife sharpness could slow down the slaughter rate and warrants extra labor and capital investment. 

The traditional method to test the knife sharpness in slaughterhouses is based on a paper test under which a piece of paper is held by one corner, and if the knife can cut the hanging-dangling part of the paper, it is considered to be sharp, and if it fails to do so, then blunt. A sharp knife (in a dry state) should slice a standard paper (A4 printer paper, 80 g weight) hanging by one corner. It is a rapid and cost-saving method but can only detect the sharpness in the center of the knife, leaving untested the other commonly used area of the knife during slaughter operations [75]. Under kosher slaughter after sharpening and in between the slaughter of animals, the shochet carefully checks the knife for roughness and nicks by running his fingernails up and down on the edge of the *Chalaf* (Figure 2). Keeping in mind the sharpness of the *Chalaf*, this should be carried out with the utmost precautions and after proper training. 

In present-day slaughterhouses with high throughput, it is quite challenging to quantify the sharpness of knives every time. ANAGO^®^, New Zealand has developed an instrument called an ANAGO^®^ sharpness tester for the objective of accurate and reliable assessment of the sharpness of knives as well as to assist in effectively controlling and improving the sharpness to the desired level. This helps in measuring knife sharpness and achieving the desired sharpness [76]. The equipment can be applied to ensure sharpness in slaughterhouses on a large scale. This instrument can also detect the presence of nicks on the blade. 

The ANAGO^®^ sharpness score presents a sharpness profile of the knife from the tip to the heel based on the relative force required to cut. The sharpness score ranges from 2.0 (42 times as much force required) to 10.0 (no force required while cutting). For measuring the knife sharpness score, sections of 20 mm blocks in length on the blade are divided from the tip to the end [76]. A sharpness score of 8.0 could be judged as in the sharp category, while most of the knives used in commercial slaughterhouses were reported to have an average of 7.80 [51]. Additionally, to maintain the sharpness level to a score of 8.0 or above, it is necessary to sharpen the knife by machine after every slaughter. In contrast, manual sharpening of the knife after every slaughter can achieve a maximum score of 7.80 [51] as presented in Figure 3. 

Figure 4 presents Chalaf used for the kosher slaughter of animals. 

## 7. Determinants of Knife Sharpness

The overall impact and outcome of the knife sharpness are affected by several factors, viz., training and expertise of the slaughterers, neck cut position, restraints, and slaughter position. These factors improve the efficiency of knife sharpness, neat-clean neck cut, and improve bleeding efficiency, thereby maximizing animal welfare compliance while alleviating pain and distress. 

### 7.1. Neck Cut Position

The neck incision under kosher slaughter is usually done at the position of cervical vertebra C2–C4. However, if done at C1, there are fewer incidences of false aneurysm formation, premature blockage of blood loss, and accumulation of blood in the upper and lower respiratory tract, later associated with unpleasant sensory signals [19]. Neck cut position is a key determinant in the onset of a false aneurysm [79] usually appearing between 7–21 s of the neck cut [80]. However, the case of a higher cut (C1) may potentially result in a cut to the larynx, including the associated bones, which may damage the knife’s sharpness and result in the rejection of the meat due to non-compliance with religious practices, particularly in Kosher slaughter [38]. Furthermore, cattle slaughtered with C1 neck cuts produced a different sound than those with the neck cut at C2–C4 [19]. Similarly, Gibson et al. [81] also noted a lower time to final collapse in anima, reducing suffering, upon performing a high neck cut. It could be due to the increased branching of carotid arteries at a higher neck position (above tracheal ring 2), thereby minimizing or preventing the retraction of carotid arteries within the connective tissue sheath [41,81]. 

The neck severance at the C1 would further reduce the potential risk of irritation associated with blood infusion into the respiratory tract due to the cutting of laryngeal nerves (transferring signals from the upper respiratory tract) and the vagus nerve (transmitting signals from the lungs and inferior trachea) [19,82]. In a controlled trial with captive bolt stunning of cattle, Gregory et al. [80] observed that making a neck cut at the C3 position had four times higher chances of early arrested blood flow and a 2.5 times higher frequency of false aneurysm formation compared to neck cutting at the C1 position. Gregory et al. [80] attributed the benefits of a sharp and clean cut at C1 using a very sharp knife with rapid blood loss and lower incidences of the formation of the false aneurysm to the following factors, viz., 

(1)Higher branching of the carotid artery at the C1 position lowers the chances of sealing all carotid artery branches.(2)The C1 neck cut needs stretching of the chin, thereby stretching the artery with less chance of their retraction within the connective tissue.(3)The presence of less connective tissue at the C1 position.

### 7.2. Skill and Training of Slaughterers

Under kosher slaughter, it is mandatory for a shochet to undergo rigorous training and education to obtain a license to slaughter animals. However, such training and skill requirements are not mandatory in halal slaughter. A trained slaughterer cuts both carotid arteries and jugular veins more effectively by making a fast swift cut close to the jaw, thereby ensuring efficient bleeding and early onset of unconsciousness. Grandin [83] observed a very short collapse time for cattle (5 s) if kosher slaughtered by a good shochet whereas a collapse time of up to 1 min in the case of a throat cut by a poor shochet. Further, a proper cut also ensures lower incidences of false aneurysms. 

During neck cutting, the average numbers of cutting movements vary with the skill of the slaughterer and the restraint system used for immobilizing the animal. Cenci-Goga et al. [84] observed variations in the number of cutting movements between halal and kosher-shackled and slaughtered sheep due to the difference in the skill of the slaughterers during neck cutting. The authors [84] in a survey on religious slaughter in Italy observed 25.2 cutting movements for halal-slaughtered cattle in the upright position, 2.9 for halal-slaughtered sheep shackled and the neck cut, and 1.25 for kosher-slaughtered sheep shackled and the neck cut. In mechanically turned restraints, fewer cutting movements were reported by the authors in the same study. The higher number of cuts directly influences the pain felt by animals during slaughter. Gregory [85] reported a sudden nociceptor discharge lasting for 4 s upon throat cut. 

Thus, proper training/technical knowledge for gentle handling of animals to handlers, less stressful restraint devices during immobilization, elimination of distractions, and appropriate neck cutting with extremely sharp knives would all maintain high animal welfare standards while producing high-quality meat with fewer instances of petechial hemorrhages [9,32].

### 7.3. Restraints and Slaughter Position

Gentle stretching of the neck during neck cutting with proper restraint improves the quality of the neck cut. The wounds should be left open to alleviate pain and facilitate bleeding. After the throat cut, the animal should be released from restraint. An inverting-conscious animal could aggravate the fear and distress in animals as well as the aspiration of ruminal fluids coupled with compression of internal thoracic organs exerting pressure and inhibiting respiration [86]. During neck cutting, the neck should be properly within reach of slaughterers for neck cutting in a swift movement. Velarde et al. [59] also observed fewer cuts (three cuts) performed during the slaughter of cattle at 90° position on their sides as compared to higher cuts (five cuts) in cattle slaughtered at 180° position on their back as well as nine cuts performed in slaughter in the upright position. 

During kosher and halal slaughter in Italy, cattle are restrained in the upright position [84]. Stressful restraint causes struggling, thereby masking the behavioral response to neck cuts. The restraints should not provide excess pressure and avoid jerky movements, which could make animals exited and agitated, thereby affecting the throat cut and slaughter process. A calm animal loses consciousness earlier than an agitated animal [83]. Animals should not be suspended by limbs except for poultry and there must not be any injury inflicted to restrain animals such as the cutting of tendons [1]. Small ruminants suspended on a shackle were reported as having significantly higher struggling as compared to manual restraints on their side [87]. Act of abusive and rough handling such as beating, poking with pointed sticks, dragging, leg clamping, shackling, and hoisting is also strictly prohibited under animal welfare and slaughter legislation [9,22].

The restraint equipment used for halal and kosher slaughter should hold the animal in a comfortable upright position before and during the slaughtering process. The same standards should be applied if a rotating box is used for restraining the animals [88]. Velarde et al. [59] observed that cattle in halal slaughter without stunning were restrained by four different methods, namely, by turning the animal at 45°, turning on their side (at 90°), turning on their back (180°), and upright positions with modified ASPCA pens. The authors [59] observed vocalization in fewer cattle in upright restraint boxes as compared to inverted restraints/turned on their backs. The cattle restrained by turning on their side lost posture early followed by cattle that were turned 45° and slaughtered in the upright position, with the longest time to loss of posture recorded in cattle turned on their back [59]. 

## 8. Prospects and Challenges

Maintaining the proper level of knife sharpness in modern slaughterhouses with higher slaughter rates is quite challenging. In the case of religious slaughter, it is required to slaughter as per the specific requirement of animal handling, knife specification, and good neck cut as the recommended practice of halal or kosher to make it safe and fit for eating to that particular community. For maintaining the desirable slaughter line speed, more infrastructure and human resources are required in religious slaughter. This will result in increasing the cost of production of meat. 

The knife size and sharpness sometimes do not match the prescribed guidelines in cases of religious slaughter. There are more incidences reported in various slaughterhouses regarding the use of shorter and poorly sharpened knives during the slaughter of animals than using the desired ones [62]. One survey conducted over ten years duration reported that over 80% of abattoir workers used an inappropriately sharpened knife [89]. The slaughterers performing repetitive tasks continuously for hours could also result in the cutting of the corners of the blades [45]. Using blunt/less sharpened knives has been associated with lower production and increased injuries due to greater force during meat cutting [90]. 

A razor-sharp knife requires lower grip force, cutting time, and cutting moments, whereas a blunt knife was projected to require 25% more cutting moments and musculoskeletal disorders (MSDs) [91]. Further, Grandin [92] described how using an inadequately sharp knife results in a higher force applied by the operator and an increase in cutting time. Further, a sharp knife is more productive by increasing the cutting speed by 1.5 times [92]. Thus, a small increment in knife sharpness could result in a significant gain in overall productivity by improving the speed of production and the safety and quality of the meat.

There is an urgent need to make people aware of the importance of knife sharpness and the role played by a sharp knife in improving and maintaining high standards, particularly in religious slaughter without stunning. There should be proper emphasis on regular training and refresher courses for the personnel involved in this meat production. A study by Claudon et al. [91] with 196 respondents in France revealed that 42% of the 196 butchers complained that the blade being used was not sharp enough and only 16% stated having been trained in sharpening and maintenance of knives. There should be national-level training programs for livestock handlers, slaughterhouse workers, and staff. The personnel employed in meat production should have a certificate of competence. 

Islam and Judaism have given utmost importance to knife sharpness, but the specific detailing of the knife is not mentioned for halal slaughter, as that mentioned in kosher slaughter. Thus, it is recommended to develop proper specifications for the knife used for halal slaughter [27]. There is a need to make slaughter workers and butchers aware of the importance of knife sharpness and its crucial role in ensuring the slaughter of animals with minimal pain. This follows the intrinsic principles of animal welfare prescribed in Islam and Jewish. 

Further, slaughterhouse management should be more concerned with proper facilities and the gentle handling of animals during slaughter. The butchers/shochet should undergo regular training or refresher course to update them about the importance of gentle handling, knife sharpness, and a neat-clean neck cut. This will protect from scotoma (factory blindness) and compassionate fatigue in slaughterers due to a monotonous work profile. There is a need for regular auditing and inspection of slaughterhouses for proper compliance with knife sharpness and other factors that affect animal welfare during slaughter. 

There is a requirement for proper monitoring of the whole process of kosher and halal slaughter. Various outcome-based measures or variables should be continuously monitored to improve the process. The high levels of vocalization (moo or bellow) during handling and restraint are associated with physiological stress due to aversive conditions such as excessive pressure from a restraint device, the sharp edge of a restraint sticking into an animal, or the use of electric goads [93,94,95]. Cattle undergoing ritual slaughtering in a Weinberg pen (in which the animal is inverted during slaughtering) were recorded to spend eight times longer time and have significantly higher cortisol and hematocrit values as compared to cattle slaughtered conventionally or in an ASPCA (American Society for Prevention of Cruelty to Animals) pen (in which the animal remains standing during slaughtering) [96]. Time to collapse, loss of posture, or eye rollback is an important indicator of animal welfare and should be continuously monitored [88]. The loss of consciousness should occur within 30–40 s after the neck cut [88]. In addition, there is a need for constant supervision and monitoring of knife sharpness by the slaughterhouse management as people tend to become sloppy when monitoring and oversight by management are reduced. 

The authors believe there should be improved knife sharpening devices, and slaughter persons should be trained in knife sharpening. The slaughter persons should have basic knowledge of animal sentience, pain, and distress associated with slaughter without stunning and various ways to alleviate the pain and distress during slaughter. Sufficient human resources with expertise in slaughter techniques and proper knowledge of religious values should be employed to reduce the work pressure in slaughterhouses. Alternatively, the slaughterhouse management may provide a set of sharp knives for slaughtering a group of animals in a day or a shift depending on usage and need. At a point of slaughter, more than one slaughterer should be appointed so that when one person is slaughtering, another will inspect and sharpen the knife and, later, vice versa. The management should also practice regular interviews/interactions with workers to assess their compassion and empathy towards animals and the intrinsic value of various guidelines prescribed in religious slaughter. Whenever needed, they should intervene accordingly. 

## 9. Conclusions

Knife sharpness plays a crucial role in rapid and clean neck severance, alleviating pain and stress in animals for producing good quality meat following religious practices. With the ever-increasing demand for halal and kosher meat due to its authenticity, nutritive value, and animal welfare compliance, it is of utmost importance to emphasize knife sharpness during slaughter as per the prescribed religious practices. Manufacturing design of slaughter knives for ergonomics, maintenance, sharpness, and assessment must be conducted for food safety and animal welfare, as well as for the sensory appeal of meat such as the palate knives and perceived healthiness.

## Figures and Tables

**Figure 1 animals-13-01751-f001:**
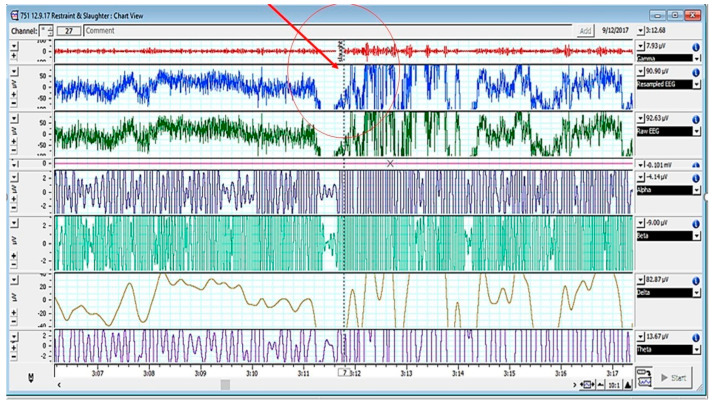
EEG spectrum of cattle pre- and post-slaughter (Note: encircled/red arrow showing the point of slaughter). Adopted from [3].

**Figure 2 animals-13-01751-f002:**
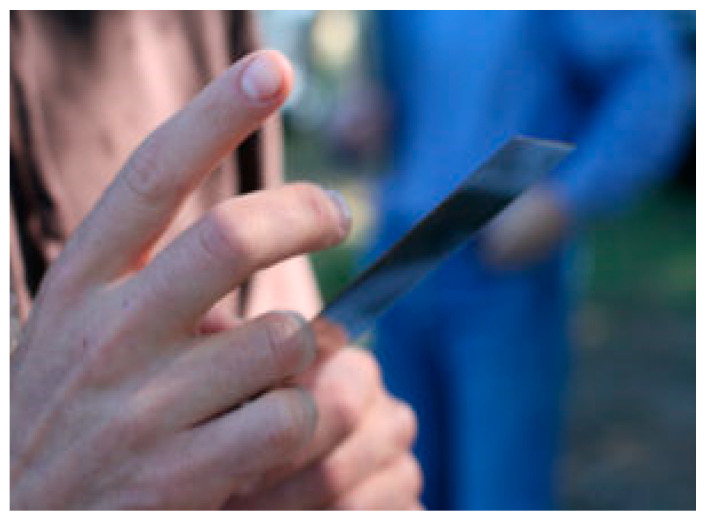
*Chalaf* sharpness test by shochet during kosher slaughter. Source: [60]. Copyright permission was obtained from the figure owner on 18 May 2023.

**Figure 3 animals-13-01751-f003:**
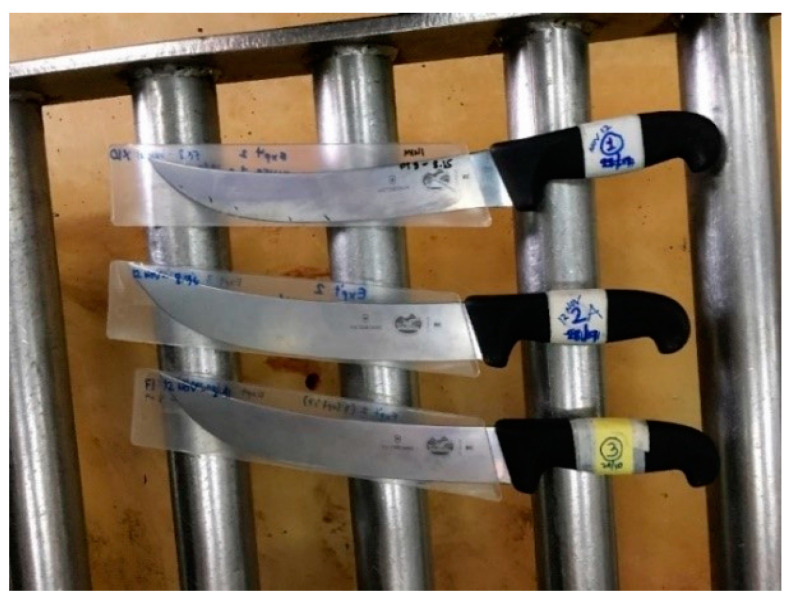
Knives used for the halal slaughter of cattle. Source: [77]. Copyright permission was obtained from the Universiti Putra Malaysia.

**Figure 4 animals-13-01751-f004:**
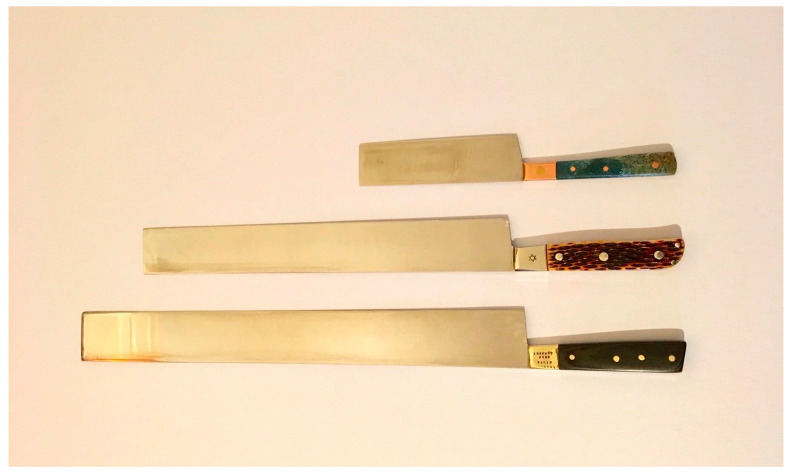
Chalaf used for the kosher slaughter of animals. Source: [78]. Copyright permission was obtained from the figure owner on 19 May 2023.

**Table 1 animals-13-01751-t001:** Blade length (approx. cm) of knife used for halal and kosher slaughtering.

Species	Halal Slaughter	Kosher Slaughter
Cattle	29.6 ± 1.79 cm (Also varies from 18 cm for cow to 24 cm for buffalo)	40.0
Poultry	–	13.5
Sheep	22.2 ± 1.82	25.4

(Source: [59,60]).

## Data Availability

Data will be made available on request.

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
