# Peer review of "Importance of Knife Sharpness during Slaughter: Shariah and Kosher Perspective and Scientific Validation"

_animals, 2023, doi:10.3390/ani13111751_

Round 1

Reviewer 1 Report

I think you should perhaps show pictures of the two kinds of knives used. There is a lot of generalization about "short " and "Long" knives . For instance a "knife should be twice as along as". There is a significant difference between cattle, sheep, goats and chickens. Perhaps a table for each, showing contrast between Halaal and Kosher knives would be useful. In South Africa,  smallstock may be decapitated on the ground, then hoisted but is sometimes hoisted, then cut.  Cattle are the main problem - manual handling of a small jersey cow and a large  meat breed bull are very different problems.  somtimes, for Halaal /Kosher slaughter, all cattle go into a sort of a crush , which is  inverted mechanically so that the throat is stretched out. 

Language problems: 

Islam and Judaism have given utmost im-                                                   Line  73

portance and zero tolerance towards animal welfare compliance d   [PCM1] 

 lower incidences of the formation of the false aneurysm [PCM2]                     Line 77

In this context This paper  critically  reviews the significance of knife sharpness in religious animal slaughter without stunning;                                                                          Line  81

under the broad ambit of animal welfare principles. By alleviating stress and pain, this  Line 82

ultimately improves meat quality[PCM3] .

in plants is based on a paper test             Line361[PCM4] 

 [PCM1]Non-compliance

 [PCM2]Lowers the incidence of a false aneurism

 [PCM3]Long sentence

 [PCM4]The word “plant” is confusing as it could also mean cutting a green plant. Maybe substitute “in slaughter houses”

See above

Author Response

The authors gratefully acknowledge the critical and in-depth observations by the reviewers. These comments have helped us in improving the quality of the manuscript. We have edited the manuscript accordingly.

Further, it is certified that all the issues raised by the reviewers have been incorporated into the revised manuscript. All the changes were marked in RED color text.

Comment: I think you should perhaps show pictures of the two kinds of knives used. There is a lot of generalization about "short " and "Long" knives . For instance a "knife should be twice as along as". There is a significant difference between cattle, sheep, goats and chickens. Perhaps a table for each, showing contrast between Halaal and Kosher knives would be useful. In South Africa, small stock may be decapitated on the ground, then hoisted but is sometimes hoisted, then cut. Cattle are the main problem -

manual handling of a small jersey cow and a large meat breed bull are very different problems. somtimes, for Halaal /Kosher slaughter, all cattle go into a sort of a crush , which is inverted mechanically so that the throat is stretched out.

Response: Thank you so much for your valuable input. We have added Table 1 comparing the blade length of knives used in halal and kosher slaughter as well as two more figures of Chalaf. We have also highlighted the points animal restraint methods including cattle restraints in the introduction (para 3) and also in the restraints section (7.3).

Comment: Language problem; Line 73, 77, 81, 82, 361, PCM2, PCM3

Response: we have gone through the whole manuscript and edited it accordingly.

Comment: The word “plant” is confusing as it could also mean cutting a green plant. Maybe substitute “in slaughter houses

Response: The plant word is replaced by slaughterhouses

Reviewer 2 Report

Line 197-198: delete the sentence as it contradicts the text presented in this section.

Line 200-203: insert “In the USA,” at the beginning of the sentence.

Line 245-248: delete ‘peaceful’ in this sentence – Temple Grandin didn’t report this.

Line 293-297: the authors should say whether neck cutting practice such as this is compatible with religious requirement. 

Section 7.2: Skill and training of slaughterers. Authors should make a clear distinction between the requirements for Kosher and Halal. In this sense, they should emphasize the fact that a shochet undergoes years of training and education to obtaining a licence to slaughter animals, whereas any practicing Muslim can perform halal slaughter without any form of training and skill test.

Section 7.3: readers would benefit if the authors could list and describe different restraining methods used for slaughter without stunning.

Line 456-457: this is not the case in Europe. Make it clear where this is practiced.

Line 461-464: make it clear to readers who are not familiar that cattle are restrained in a rotating pen restraint. In this restraining method, cattle may be restrained in an upright position or rotated to 90 or 180o prior to neck cutting.

Line 465-466: In which country / continent? This method is illegal in Europe.

Section 8: emphasize the importance of certificate of competence and the need for staff training and certification programmes at national levels.

Line 507: delete ‘no or’

Line 517-518: not just the sharpening of knife, they should also acquire the basic knowledge of animal sentience, pain and distress associated with slaughter without stunning, and various ways of minimising pain and suffering in animals as highlighted in this paper.

Author Response

The authors gratefully acknowledge the critical and in-depth observations by the reviewers. These comments have helped us in improving the quality of the manuscript. We have edited the manuscript accordingly.

Further, it is certified that all the issues raised by the reviewers have been incorporated into the revised manuscript. All the changes were marked in RED color text

Comment: Line 197-198: delete the sentence as it contradicts the text presented in this section

Response: deleted as per suggestions

Comment: Line 200-203: insert “In the USA,” at the beginning of the sentence

Response: Edited

Comment: Line 245-248: delete ‘peaceful’ in this sentence – Temple Grandindidn’t report this.

Response: Thank you so much for pointing  this. We have edited the sentence as per cited reference.

Comment: Line 293-297: the authors should say whether neck cutting practicesuch as this is compatible with religious requirement

Response: Thank you so much for the observation. We have added additional sentence highlighting the need for its approval and guidelines.

Comment: Section 7.2: Skill and training of slaughterers. Authors should make a clear distinction between the requirements for Kosher and Halal.In this sense, they should emphasize the fact that a shochet undergoes years of training and education to obtaining a licence to slaughter animals, whereas any practicing Muslim can perform halal slaughter without any form of training and skill test.

Response:  Added as per suggestion

Comment: Section 7.3: readers would benefit if the authors could list anddescribe different restraining methods used for slaughter without stunning.

Response: we have added 2 para on this topic as suggested by the reviewer.

Comment: Line 456-457: this is not the case in Europe. Make it clear where this is practiced.

Response: The sentence has been deleted. This is majority practised in Asia and Africa.

Comment: Line 461-464: make it clear to readers who are not familiar that cattle are restrained in a rotating pen restraint. In this restraining method, cattle may be restrained in an upright position or rotated to 90 or 180o prior to neck cutting.

Response: We have added one separate para on animal restraining.

Comment:Line 465-466: In which country / continent? This method is illegal inEurope.

Response: The sentence is edited and stuck after shackling has been deleted. We are in agreement with the reviewer observation as in the EU directives it is prohibited.

Comment: Section 8: emphasize the importance of certificate of competenceand the need for staff training and certification programmes at national levels

Response: Added as per suggestion

Comment: Line 507: delete ‘no or’

Response: Edited

Comment: Line 517-518: not just the sharpening of knife, they should also acquire the basic knowledge of animal sentience, pain and distress associated with slaughter without stunning, and various ways of minimising pain and suffering in animals as highlighted in this paper.

Response: Added as per suggestion

Reviewer 3 Report

This paper covers the importance of using a very sharp knife when slaughter is performed without stunning.  This will help reduce suffering.

Line 62 - Add a paragraph to further emphasize the importance of using less stressful methods of restraint.  Some of the worst animal welfare issues this reviewer has observed with kosher or halal slaughter were associated with highly stressful methods of restraint and handling.

Line 195 - In very cold climates, washing animals as they center the lairage would cause severe cold stress.  In many parts of the world that have cold weather, the lairage is NOT climate controlled.

Line 218 - You may want to include two references that show that sheep lose consciousness faster than cattle.  This is due to differences in their anatomy.

Baldwin, B.A. and Bell, F.R. (1963) The anatomy of the cerebral circulation of the sheep and the ox: The dynamic distribution of the blood supplied by the carotid and vertebral arteries to cranial regions.  J. of Anatomy, 97:203-215.

Line 304 - Provide a reference for the NAGO Sharpness score.

Line 335 - Change the word "superior" to "greater."

Line 362 - Change "free swimming" to the "hanging dangling."

Line 516 - There needs to be an additional section that explains that the entire kosher or halal slaughter process needs to be constantly monitored and measured.  This will make it possible to have continuous improvement. Some of the variables that should be measured are:

- Vocalization of cattle during restraint - High levels of vocalization (moo or bellow) during restraint are associated with physiological measures of stress.  Dunn, C.S. (1990) Stress reactions of cattle undergoing ritual slaughter with two methods of restraint, Veterinary Record, 126, 522-525. Improved handling and restraint will reduce the percentage of cattle that vocalize.

- Time to collapse or eye rollback - Welfare of the animal is improved if loss of consciousness occurs more quickly. This measure is really important for continuous improvement.

- Continuous monitoring of knife sharpness - To keep standards high requires continuous supervision and monitoring.  People tend to get sloppy when monitoring and oversight by management is reduced.

Fine

Author Response

The authors gratefully acknowledge the critical and in-depth observations by the reviewers. These comments have helped us in improving the quality of the manuscript. We have edited the manuscript accordingly.

Further, it is certified that all the issues raised by the reviewers have been incorporated into the revised manuscript. All the changes were marked in RED color text

Comment: This paper covers the importance of using a very sharp knife when slaughter is performed without stunning. This will help reduce suffering.

Response: Thank you so much for your positive observations.

Comment: Add a paragraph to further emphasize the importance of using less stressful methods of restraint. Some of the worst animal welfare issues this reviewer has observed with kosher or halal slaughter were associated with highly stressful methods of restraint and handling.

Response: Information of the importance of less stressful restraint has been added as per suggestions.

Comment: Line 195 - In very cold climates, washing animals as they center the lairage would cause severe cold stress. In many parts of theworld that have cold weather, the lairage is NOT climate controlled

Response: Sentence edited as per suggestions.

Comment: Line 218 - You may want to include two references that show that sheep lose consciousness faster than cattle. This is due to differences in their anatomy - Baldwin, B.A. and Bell, F.R. (1963) The anatomy of the cerebral circulation of the sheep and the ox: The dynamic distribution of the blood supplied by the carotid and vertebral arteries to cranial regions. J. of Anatomy, 97:203-215.

Response: Thank you so much for your valuable observation. We have added AVMA and Baldwin and Bell references in restraint importance section in Introduction.

Comment: Line 304 - Provide a reference for the NAGO Sharpness score.

Response: Added as per suggestion

Comment: Line 335 - Change the word "superior" to "greater."

Response: Edited

Comment: Line 362 - Change "free swimming" to the "hanging dangling."

Response: Edited as per suggestion

Comment: Line 516 - There needs to be an additional section that explains that the entire kosher or halal slaughter process needs to be constantly monitored and measured. This will make it possible to have continuous improvement. -------------------------------------------------------- monitoring and oversight by management is reduced.

Response: Thank you so much for your valuable suggestions. We have added separate para on the suggested topics.
